# Development and Validation of a Physical Activity Educational Module for Overweight and Obese Adolescents: CERGAS Programme

**DOI:** 10.3390/ijerph16091506

**Published:** 2019-04-28

**Authors:** Xiao Chuan Lau, Yaw Loong Wong, Jyh Eiin Wong, Denise Koh, Razalee Sedek, Ahmad Taufik Jamil, Alvin Lai Oon Ng, Abu Saad Hazizi, Abd. Talib Ruzita, Bee Koon Poh

**Affiliations:** 1Nutritional Sciences Programme & Centre for Community Health, Faculty of Health Sciences, Universiti Kebangsaan Malaysia, 50300 Kuala Lumpur, Malaysia; lxc1220@gmail.com (X.C.L.); wongyl1224@gmail.com (Y.L.W.); wjeiin@ukm.edu.my (J.E.W.); rzt@ukm.edu.my (A.T.R.); 2Centre for Education and Community Well-being, Faculty of Education, Universiti Kebangsaan Malaysia, 43600 Bangi, Selangor, Malaysia; denise.koh@ukm.edu.my; 3Centre for Biotechnology and Functional Food, Faculty of Science and Technology, Universiti Kebangsaan Malaysia, 43600 Bangi, Selangor, Malaysia; razalee@ukm.edu.my; 4Department of Public Health Medicine, Faculty of Medicine, Universiti Teknologi MARA, 47000 Sungai Buloh, Selangor, Malaysia; taufik70@uitm.edu.my; 5Department of Psychology, School of Science and Technology, Sunway University, 47500 Petaling Jaya, Selangor, Malaysia; alvinn@sunway.edu.my; 6Department of Nutrition and Dietetics, Faculty of Medicine and Health Sciences, Universiti Putra Malaysia, 43400 Serdang, Selangor, Malaysia; hazizi@upm.edu.my

**Keywords:** physical activity, overweight, obesity, adolescents, educational module

## Abstract

Educational modules can be effective in educating and motivating adolescents to participate in physical activity (PA). This paper describes the development and validation of a PA educational module for use in an obesity intervention programme, CEria Respek Gigih Aktif Sihat (CERGAS). The present study was conducted in three phases: Phase I was composed of needs assessments with four focus group discussions to elicit adolescents’ opinions regarding module content and design, Phase II was the development of the PA module, while Phase III was content and face validation. A content validity index (CVI) was used to assess content validity quantitatively, with a CVI of more than 0.79 being considered appropriate. The needs assessments indicated that content of interest to adolescents included: the benefits of exercise; exercise techniques; ways to increase PA and how to stay motivated. Attractive graphic design was determined as a way to draw the adolescents’ attention. The module covered five topics: “Let’s Be Active”; “Exercise and Fitness”; “Staying Safe during Exercise and Physical Activity”; “Planning for Success” and “How to Overcome Sedentariness and Get Moving”. The module was found to have good content validity (mean CVI = 0.85). Expert members made suggestions to improve the module. These suggestions were then accepted, and the module was modified accordingly. We concluded that the module has good content validity and can be used to improve PA knowledge amongst CERGAS participants.

## 1. Introduction

Over the past decade, the leading causes of death worldwide have shifted from infectious to chronic diseases. Strong evidence indicates that lack of physical activity (PA) is an important predictor of some chronic diseases, including obesity and cardiovascular disease [1,2]. Moreover, a few longitudinal studies have indicated that low PA levels during adolescence are associated with the development of chronic diseases during adulthood [3,4]. These studies suggest that increasing PA, improving fitness and lowering weight, can benefit future health status.

In Malaysia, the prevalence of obesity among children and adolescents (<18 years old) increased from 5.4% in 2006 to 6.1% in 2011 and 11.9% in 2015 [5,6,7]. Specifically, 14.4% of adolescents aged 10 to 14 years old were obese [7]. Physical activity has long been recognised as an important factor that can be modified to prevent obesity in young people [8]. Recent studies conducted in Malaysia have shown that obese children [9] and adolescents [10] are less physically active than their normal-weight peers. A nation-wide survey of Malaysian school children has also reported that the children were generally sedentary [11]. All these studies suggest that, to prevent the increasing prevalence of overweightness and obesity over the long term, intervention to increase physical activity in schools and the community should be implemented.

Intervention programmes that focus on physical activity [12] have been known to improve the body composition of overweight and obese adolescents. Hence, the present study developed a combined PA and nutrition intervention, namely, the CERGAS programme to enhance knowledge of PA and to encourage PA among adolescents. The CERGAS is the Malay abbreviation of CEria (cheerful); Respek (respect); Gigih (persistence); Aktif (active) and Sihat (healthy). It is a 12-week programme where participants receive additional nutrition and PA education delivered by health professionals in a two-day residential education camp over a weekend, apart from the standard Physical and Health Education school curriculum. Participants also take part in exercise training at school, twice weekly for 12 weeks, for 90 min per session, after regular school hours.

Interventions aimed at improving adolescents’ physical activity behaviours have focused on a number of contexts within which adolescents spend their time, including at home [13], school [14], or community-based organizations [15]. Less attention has been paid to the potential of providing training at residential education camps. There are several reasons why CERGAS employed an education camp setting to improve adolescents’ PA behaviours. First, a hallmark feature of education camps is the provision of developmentally appropriate support and opportunities aimed at fostering skill building and personal growth for adolescents [16]. For many adolescents, the camps provide supportive peer and adult relationships and new resources and experiences not available at home or in their school environments. Second, camps are sustained and immersive experiences that have the intensity and duration to have a substantial impact on adolescents’ developmental outcomes [17]. Third, the frequency of school visits can be minimized. This will ensure that the adolescents are able to focus on their academic activities without too many disruptions during school hours. Further, camps offer participants the opportunity to immediately apply and practice the knowledge and skills learned during the camp itself.

Previous studies have reported that Malaysian children and adolescents scored low on PA knowledge [18,19]. The low knowledge scores could be related to the lack of prior exposure to a PA education programme. An effective channel where secondary school adolescents can get exposure to such knowledge is during physical education (PE) and health education classes. School PE curriculum incorporates basic and general PA knowledge; however, ironically, most PE teachers do not use the PE period to deliver the theory parts [20,21]. This could be explained by the lack of qualified PE teachers and pedagogical knowledge [20]. It has also been reported that the time allocated for PE class is insufficient to teach the syllabus designed for the subject [21]. By taking these limitations into consideration, CERGAS PA education is delivered by health professionals in a two-day education residential camp conducted over a weekend. In future, there is need for developing partnerships with the Ministry of Education to develop and implement PA intervention.

Education module appears to be effective in educating and motivating children and adolescents to be involved in physical activity [22]. Previous interventions that used teaching strategies (e.g., CATCH (Child and Adolescent Trial for Cardiovascular Health), SPARK (Sports, Play, and Active Recreation for Kids), M-SPAN (Middle School Physical Activity and Nutrition), Move it Groove it) [23,24,25,26,27] had effectively increased moderate-to-vigorous PA (MVPA) during PE lessons. For example, results from the CATCH intervention increased MVPA during PE lessons by 12% to meet the current recommendation for children’s PA during PE lessons [27]. In Malaysia, however, there is a lack of PA education modules targeting young adolescents. The existing education modules developed by the Ministry of Health Malaysia and local studies aiming to tackle overweightness and obesity among children and adolescents mostly focus on changes in dietary intake and habits [28,29,30]. Thus, there is a need for an education module that focuses on PA in order to overcome the trend of adolescents becoming overweight and obese. The present study aimed to develop and validate a PA education module for overweight and obese (O/O) adolescents. This PA education module will serve as the main education material for CERGAS’ participants and facilitators in the two-day education camp conducted at a training centre.

## 2. Materials and Methods

### 2.1. Study Design

The present study was conducted in three phases, namely, Phase I: Needs Assessment, Phase II: Development of PA module and Phase III: Content and Face Validation of PA module. Ethics approval was obtained from the Research Ethics Committee of Universiti Kebangsaan Malaysia (Project Code: NN-103-2012). Permission from the Ministry of Education and the Kuala Lumpur Federal Territory Education Department was obtained prior to the commencement of data collection. Only participants with written, informed consent from their parents or legal guardian were recruited. The validated PA module can serve as reference material to guide secondary school teachers once the module has been proven effective in improving secondary school adolescents’ knowledge, attitude and practice (KAP). The content of the CERGAS PA education module can be included into school physical education (PE) textbooks to improve students’ PA KAP. The effectiveness of the validated module in improving adolescents’ KAP was evaluated prior to implementation.

### 2.2. Phase I: Needs Assessment

The CERGAS programme employed an “adolescent-centred” approach. Thus, focus group discussions (FGDs) were used to explore the opinions, ideas, perceptions and concerns of adolescent participants regarding the content, layout and design of the module. The FGD questions were scripted accordingly. Four FGD sessions were conducted amongst Form One to Form Three students, aged 13 to 15 years, in two secondary schools in Kuala Lumpur. The age, sex and weight status of the participants were taken into consideration when organizing the focus groups. Obese, overweight and normal weight adolescents were included in the study to learn what the participants understood about obesity and the behaviours related to obesity. Each session had a maximum of 10 students with different weight status (normal weight or O/O). In addition, each session had only male or only female participants.

The questions followed a natural form of conversation and were asked in a structured sequence. Most of the questions were open-ended to encourage the participants to provide more ideas related to the content, design and format of educational modules that will attract adolescents’ attention. The main questions were:
(i)What topic do you want to know more about PA or exercise?(ii)In your opinion, what topics should be included in PA and exercise educational materials?(iii)In your opinion, what are the factors that cause you to be not interested in reading educational materials?(iv)In your opinion, what kind of educational design will interest you to continue reading?(v)In your opinion, what kind of educational materials format will attract/encourage you to read?


At the end of the session, participants were asked “*Is there anything else you would like to share with us*”, to allow them to provide information inadvertently omitted during the FGDs. This FGD interview protocol was reviewed by expert panels comprising dietitians, nutritionists and qualitative research expert.

All FGD sessions were conducted in a private room at the respective schools to avoid outside noise and distractions. During the sessions, an MP3 recorder (NWZ-B172F) was used to record the conversations. The facilitators took field notes and noted the participants’ non-verbal responses, such as their tone of voice, body language and facial expressions. Each session took about 50 min to one hour. Before FGD sessions ended, participants were given the opportunity to provide brief personal commentaries by writing on a piece of note paper. Researcher collected the notes and incorporated the responses into module preparation. The sessions continued until “saturation” point, that is, the point when the participants produced little or no new information. We transcribed the focus group data using an intelligent verbatim style, where the fillers, background noises and repetitions were omitted. Respondent validation was done by referring to the participant. The participants verified the transcripts and confirmed the information was consistent with the opinions they had shared during the sessions. Field notes on non-verbal cues, such as body language or facial expressions, were incorporated in the transcripts after completion of respondent validation.

The researchers read the transcripts several times until a sense of the whole was obtained [31]. After making sense of the data, inductive analysis was conducted. Important points reflecting the main interest of the study were highlighted. The notes and headings were written in the text. After that, the written material was reviewed again, and the relevant headings were written down in the margins. Thereafter, all the headings were collated and transferred on to coding sheets. Subsequently, codes were assigned to the headings that expressed the same concept. The related codes were then grouped into sub-categories. Sub-categories with similar events were grouped together as categories. Finally, the themes were identified, and each theme was named using content-characteristic descriptions. These themes explained the topic being studied [31]. During the data analysis process, the researchers reviewed and refined the themes. The themes and quotes were then translated from Malay to English to ensure that the actual meaning had been retained.

### 2.3. Phase II: Development of the PA Module

#### 2.3.1. Module Structure

The intervention development committee carefully reviewed and discussed the module content. Module development involved an iterative design and review process accomplished through face-to-face meetings with experts comprising five nutritionists (X.C.L; Y.L.W; J.E.W; H.A.S; and B.K.P), a psychologist (A.N.L.O), a public health medicine specialist (A.T.J), two physical education specialists (D.K and R.S) and a dietitian (R.A.T). The PA module was structured to provide a systematic guideline for facilitators to conduct the education programme to CERGAS participants. The module was written in the Malay language. The order of units in the PA module began with basic PA topics and moved on to more specific PA topics.

Layman’s terms were used in order to communicate effectively with readers. Realistic and culturally appropriate examples, attractive images and eye-catching design and colour were used. To encourage participants to apply what they had learnt, interactive learning activities, such as quizzes, group discussion or interactive homework, were incorporated at the end of the units in each module. As the participants’ parents and school teachers would use the PA education module as reference, our module was developed using appropriate language in an easy-to-understand format.

#### 2.3.2. Module Content

The intervention development committee reviewed relevant literature and PA modules published by the Malaysian Health Promotion Board [32] and international studies [33,34] to develop content for the present module. Information obtained from the FGDs was also incorporated into the module.

As Social Cognitive Theory [35] was used to design the PA module, we considered the following: (1) personal factors, such as outcome expectations and self-efficacy; (2) behavioural factors, such as behavioural capability and self-regulation/self-control; and (3) environmental factors, such as observational learning/modelling and social environment. For example, considering personal factors enabled us to incorporate information regarding correct exercise techniques in the PA module, which, in turn, would increase adolescents’ self-efficacy to exercise. In order to encourage long-term behavioural changes in PA, the CERGAS programme incorporated elements that can increase self-efficacy, behavioural competency and social support. We hypothesized that, following the implementation of the CERGAS PA education module, the adolescents’ personal factors, such as increased PA knowledge and positive attitude along with positive reinforcements from environmental factors, for instance, social support from peers and facilitators, will motivate the adolescents as well as increase their abilities to engage in physical activities by themselves.

The pedagogical approach used to implement the PA module was constructivism teaching. This method of teaching helps participants to better relate the information learned in the education camp to their lives. In a constructivism classroom, participants work in groups. This helps participants learn social skills, support each other’s learning processes and value each other’s opinions and inputs.

### 2.4. Phase III: Content Validation and Face Validation of PA Module

#### 2.4.1. Content and Face Validation by Expert Panels

A panel of experts evaluated the completed PA module for appropriateness and relevance of content. It has been recommended that there should be at least five members on the panel to avoid chance agreement [36]. Our interdisciplinary panel was composed of two nutritionists, a dietitian, a health promotion expert and a physical education specialist to enhance the exchange of ideas. The experts on our panels were deliberately selected so that they would produce diverse ideas from their different specializations [37].

Content validation was an independent review process using a content validation form. The validation form we used was adapted from an instrument proposed by Castro and colleagues [38]. The PA module was evaluated in seven aspects. Two aspects (i.e., scientific accuracy and content) related to content validity. The other five (i.e., literary presentation, illustrations, sufficiently specific and understandable material, legibility and printing characteristics, and quality information) related to face validity.

In reports of instrument development, the most extensively used approach for content validity is the content validity index (CVI) [39]. The experts examined each unit and rated the components on a scale (1 for “not relevant”; 2 for “some revision required”; 3 for “relevant but needs minor revision” and 4 for “very relevant”). To obtain the CVI, the number of experts judging an item as relevant (that is, rating 3 or 4) was divided by the total number of experts. The CVI expresses the proportion of agreement on the relevancy of each item, which is between zero and one [40]. It has been proposed that an index of 0.80 or higher is required before an item is accepted [41]. Judgment on each item was made as follows: if the CVI was higher than 0.79, the item was deemed appropriate; if the CVI was between 0.70 and 0.79, the item needed revision; and if the CVI was less than 0.70, the item was eliminated [42]. For face validation, the items with at least 75% positive responses were deemed validated [43].

#### 2.4.2. Face Validation by Target Audience

The instrument aimed at the target audience was the same as that presented to the experts but without content validity aspects. The instrument for target audience consisted of 34 items and focused only on literary presentation, illustrations, sufficiently specific and understandable material, legibility and printing characteristics, and quality information, indicating “yes” or “no”. A total of 24 secondary school adolescents participated in face validation. In addition, three small group discussions were conducted to obtain direct feedback. Involving representatives of the target group in the validation process ensures that the population for whom the instrument is being developed is represented in evaluation [44]. The criteria for inclusion were Malay secondary school adolescents who are literate in the Malay language. These adolescents were asked to read the module and to indicate the vocabulary they found difficult and to suggest alternative vocabulary which were more readily understandable. They were also asked to evaluate the adequacy of the illustrations. The validation process conducted by the experts and adolescents was carried out until there were no more recommendations for changes.

## 3. Results

### 3.1. Socio-Demographic Characteristics

A total of 44 adolescents aged 13 to 15 years old from secondary schools in Kuala Lumpur participated in the FGDs. The mean age of the participants was 14.3 (1.0) years. Table 1 shows the demographic characteristics of the participants involved in the FGDs. All the participants were Malay adolescents. Total 40.9% participants were of normal weight, while 18.2% were overweight and 29.5% were obese, and another 11.4% were underweight.

### 3.2. Description of Qualitative Findings

#### 3.2.1. Physical Activity Topics of Interest

The themes identified from the interviews were organized into two topics: (a) PA topics of interest and (b) design of education materials. Table 2 shows the summary of the main themes for qualitative findings.

The present study found that most participants were keen to know more about the correct posture or techniques to perform PA and exercise, including the advantages and disadvantages, and types of PA that would improve muscle and cardiovascular endurance. In addition, the participants wanted tips on how to get motivated to exercise and do PA regularly. The participants’ knowledge of PA and exercise was limited. For example, a participant could not provide an example of appropriate activity to improve muscular and cardiovascular endurance. In addition, a majority of the participants were not exposed to in-depth knowledge of PA. For example, while they recognized the Malaysian Food Pyramid, they had never been exposed to the PA Pyramid. Hence, broader and more in-depth discussion of certain PA topics should be incorporated into the educational materials.

Some participants provided short or single word answers which were not helpful. Male participants were more interested in improving muscular and cardiovascular endurance. Female participants were more interested in learning how exercise/PA would help them reduce weight, how to increase PA, and tips to keep motivated. Other exercise/PA-related interest topics and themes that were mentioned in the focus groups were the frequency, type, duration and benefits of exercise.

#### 3.2.2. Design of Education Materials

The FGD participants also made suggestions regarding the design of education materials. The five main themes identified were: (i) organisation; (ii) layout and typography; (iii) content; (iv) illustration and cover; and (v) language.

##### Organization

The manner in which written health education material is organized can greatly affect the extent to which the information is attended to, comprehended and retained. The use of bullet points is advocated by a number of participants. (*G4: Do not put too many points on a page, maximum five points is enough.*) According to the participants, bulleted lists are remembered and understood better than paragraphs. One participant mentioned that usually students do not like long paragraphs which would discourage them from reading further. Some participants suggested providing information in mind map forms as they were more familiar with this strategic technique (*B7: We do not prefer content in paragraph form, can do in mind map form, for instance, we use mind map to explain exercise, then elaborate advantages and disadvantages of active lifestyle.*) (*B8: Easier to memorize if do in mind map form, it is simple, we always do that during revision for exam preparation.)*

##### Layout and typography

The readers’ comprehension was influenced by the arrangement and organization of the text. Careful design of materials’ layout and typography can help readers understand easily. All the participants recommended that text should not be too long. (*P5: …wordy and lengthy poster always stop people from reading, people feel bored, long sentences make us feel confused.*) To ease comprehension of the content, shorter line length was encouraged. (*G10: The most important thing is to reduce the wordings.*) One student mentioned that as they already have too much homework, they only have limited time for extra reading. (*P1: Not interested to read those materials because no time and there are a lot of homework to complete…feel burdened to read lengthy materials…we may try to read if the materials are simple and short.*) A student expressed concern about font size and indicated a preference for larger font size. (*P5: …try to make the font size bigger a bit…*)

##### Content

From the discussions, it was clear that the objectives of the materials had to be clearly stated as readers might not pay attention to the material if they did not understand the purpose of the material. (*P7: We do not know the importance of reading those materials, we will only do so if teachers ask us to read.*) Important points should be highlighted so that readers can readily identify pertinent information. A student stated “*… if I read a poster, I always search for important information…*”. Several participants reported that the content of the materials should be relevant and offer benefits. (*P3: Not interested if the material is not related to our problems…I will read if the material has benefits for us…if read just for the sake of reading, I feel bored…*) In addition, participants also suggested using examples that are more relevant to daily life to enhance their understanding of a topic. Most of the participants also agreed that incorporating game elements (e.g., puzzles, riddles) in the materials can help them process information quicker. (*P2: Making learning fun motivates us and helps us pay attention to the subject.*) The content presented should also be more interactive to prevent boredom.

##### Illustration and cover

Illustrations should be used to improve understanding of essential information. The participants frequently reported valuing illustrations as they make the materials more attractive, and therefore, more likely to be read. Each drawing, table or figure should communicate simple and readily comprehensible ideas. (*G2: Try to make the drawings, illustrations and tables simple, don’t be too complicated, sometimes if the content is too long…will make me “headache”…*) One student added: “*if possible, put illustrations on every page* (other students nodded in agreement).” It is important that the cover of the education material is appealing and attracts attention. A student said: “The *cover of the booklet has to have attractive illustrations, so people will be attracted to read*” (*P2*). Readers’ attention can be gained if the materials are colourful, and different from school textbooks. (*P6: If possible, use colour paper, if white colour it looks common and similar to school textbooks.*) In terms of colour selection, participants preferred brighter colours. (*P7: Darker colour looks dull, if the material is designed with brighter colour, it will be more attractive, surely many people will like to read.*)

##### Language

As the written health education materials will be read by people with a wide range of literacy skills, it should be written simply, at the lowest possible reading level that conveys the information accurately. The participants preferred simple rather than complex written materials. (*P5: Try to make the written materials simple and concise.*) Using simpler terms can help with comprehension, as a student revealed: “*Can use simple language…usually most of the content written in health education materials is difficult to understand*.” Malay-speaking participants showed a strong preference for education materials in the Malay language. They felt it would be easier to learn in their native language. A participant mentioned that students are usually too lazy to check the dictionary for English words they do not know. (*P3: If written in English, we need to spend time to look up the dictionary, it’s tedious to do that…*) Another student added: “*Most students ignore English-written materials, they do not understand…to attract me to read, must be in Malay*…”. The present data reconfirm that a language barrier is one of the reasons for students’ ignorance of basic health information.

### 3.3. Module Content

The PA education module was developed based on information obtained from the FGDs and in accordance with recommendations for conception and efficacy of educational tools, referring to content, language, organization, layout, illustration, learning and motivation. The dimensions of the final version of the PA module are 8.27 inches × 11.69 inches. The PA module has 31 pages, with front cover, back cover, table of content and a page for notes. The module contains a total of five units. The topics, learning outcomes, content and activities of the module are summarized in Table 3.

### 3.4. Content and Face Validation by Expert Panel

“Scientific Accuracy” obtained a CVI of 0.80 and “Content” obtained a CVI of 0.90 for the PA module, indicating an excellent level of agreement between the experts for content validity (Table 4). As the overall CVI of the module was 0.85, the module was deemed validated for content. At the end of the validation rating, the experts were asked to provide a general opinion about the module. The module was then revised based on their recommendations. Other aspects of evaluation such as the general formatting and paragraphing, sentence structure, learning activities, and duration proposed for the training module yielded positive feedback from the expert panel members.

Although the overall CVI proved to be good (0.85), the evaluative aspect of the module, the “Scientific Accuracy”, obtained a borderline CVI (0.80). Among the evaluated items in this aspect (if the content agreed with current knowledge, whether the guidelines/recommendations presented were needed and correctly addressed) one of the five experts partially agreed with the items. The experts’ suggestions were analysed and corrections were made.

The content validation of the PA module required no major corrections, only minor amendments, such as wording and grammar. The module fulfilled the requirement of a “module” and can be used as a guide for facilitators to conduct education programmes. We propose that the module is used as suitable reference material for adolescents, parents and school teachers.

To obtain face validity of the module, the level of agreement of the expert panel members was calculated for the five evaluation aspects of the instrument (literacy presentation, illustrations, sufficiently specific and comprehensive material, legibility as well as printing characteristics and quality of information). According to Figure 1, the level of agreement among the experts was high, ranging from 75.6% to 86.7%, which was higher than the established minimum of 75% [44]. This indicates there is adequate face validity for the module. The experts also made suggestions to improve the module, both in its appearance and content, such as: changing the module title; replacing or excluding technical terms; reformulating illustrations; simplifying and restating phrases, among others.

### 3.5. Face Validation by Target Audience

As shown in Figure 2, all five face validity aspects achieved a level of agreement higher than the required minimum to be valid (75%) [43], indicating an excellent level of agreement among adolescents. The PA module was considered to have achieved face validity for the target audience. Among the 49 items evaluated by the 24 adolescents, only 11.9% responses were marked “No”, ratifying the level of acceptance and responses during the assessment of the module.

Three small group discussions were held at two secondary schools with 24 adolescents to obtain their views on both modules. The mean age of the participants was 13.8 ± 0.8 years old. The purpose of the small group discussions was to further elucidate the adolescents’ views on the design, graphics and content of the PA and nutrition modules. Certain terms that were considered difficult to be understood by the participants were discussed during the sessions to capture their level of understanding. Subjects were asked which part of the module were most attractive. From the discussion, we found that most of the adolescents preferred cartoon and colourful images, as well as information presented in the form of mind maps or diagrams.

In addition, the adolescents provided insights as to which parts of the module were unattractive and should be modified. Some parts of the module were considered too wordy as they had no illustrations. Some font sizes were too small and some of the colours in diagrams looked dull and boring. This feedback was taken into consideration. Apart from these issues, most participants claimed that they understood most of the terms used in the module because the terms were commonly used in their daily lives. However, a few terms such as “cross training” and “resistance” required explanation from the facilitators.

In terms of concept comprehension, most participants understood the concepts in the module. These included “intensity level”, “FITT formula” and “PA pyramid”. The majority of the participants understood the concepts without additional explanation from facilitators. However, all the participants asked the facilitators to explain the concept of “cross training”. The diagrams in the module facilitated the participants’ understanding of the concepts effectively and systematically. A majority of the participants understood the arrangement of the diagrams. Moreover, the participants were able to explain the messages that the content was meant to convey. This indicated that the content is appropriate for the level of comprehension of the targeted adolescent group.

The target population also evaluated the module positively. They considered the module important for the promotion of knowledge, with rich content, clear and appropriate formatting and attractive illustrations. In addition, the participants found the module relevant in supporting psychosocial aspects of their lives, such as improving their quality of life. Validating the educational material with representatives of the target audience is a necessary and important step for the researchers. It is the moment in which there is a realization about what is lacking, what has not been understood, and the distance between what is written, what is understood and how it is understood [45]. With the development and validation of the module completed, the module will undergo continuous updates. The validated PA module will be used in the CERGAS education programme. A pilot study has been conducted to determine the effectiveness of the PA module and will be reported in other paper.

## 4. Discussion

The present study successfully developed a validated PA education module. The mean CVI was 0.85, thus confirming content validity. Other studies that validated printed educational materials also used the CVI to measure content validity and went through adjustments until the validated final version was reached, demonstrating the importance of this step for the development of quality educational materials [43,46]. This process of adapting educational materials to the suggestions of experts is an essential step to make the module more scientifically rigorous and effective for health education activities [46,47]. A few studies have shown that the main resources for obtaining health-related information were television/radio, newspaper/magazine or the internet [48]. As this data is often based on evidence that is not scientific, it can mislead adolescents. Existing textbooks of physical and health education only partially cover information regarding PA and are teacher-centric. Our PA module was developed to remedy the weaknesses of mass media health-related information and textbooks.

One possible explanation for the limited effectiveness of PA promotion is a lack of awareness of health behaviour [49]. People may not be aware of what healthy levels are. In order to accurately estimate the PA levels, they need to determine their activity intensity during different behaviours, quantify their activity of this intensity, and sum it over time. In addition, the importance of PA and exercise to health and academic performance are described in our PA module. The influence of PA on weight and sleep quality was also emphasised so that readers understand how PA affects their quality of life. The PA Pyramid was also incorporated to show how much PA is required to lead a healthy and active lifestyle. Increased awareness may bring about behaviour change [50]. Improving PA awareness may therefore be a crucial initial component of promotion campaigns, although few interventions consider this [51,52].

The principle of FITT (Frequency, Intensity, Type and Time) was introduced as an exercise formula. Findings from the FGDs indicated there were students who wanted to learn how much and what kind of exercise they should do to build up muscles and stay healthy. Readers can refer to this formula when exercising. The exercises corresponding to different fitness components were also included in this module. Among the fitness components cited in our PA module were cardio-respiratory endurance, body composition, muscle strength, muscle endurance and flexibility. Appropriate examples were included to facilitate understanding. At the same time, the correct exercise sequence, techniques as well as duration were included. A study reported that one of the barriers to exercise among adolescents was lack of knowledge of PA [31]. Knowledge of PA may increase engagement in as well as adherence to regular PA routine in this population.

Participation in sport activities is a major cause of unintentional injury among Asian adolescents [53]. Bazelmans and colleagues [54] reported that obesity was associated with injuries among adolescents. Hence, appropriate injury prevention strategies, such as warm-up and cooling down before and after exercise, as well as the importance of hydration, were incorporated in this module. Subjects were taught tips to plan regular PA and exercise. Exercise or PA planning steps were included in this PA module. The PA module also included information and tips that encourage subjects to reduce sedentary behaviour and increase PA level. This topic is deemed necessary given the findings from the Malaysian School-Based Nutrition Survey 2012 [55] and Nutrition Survey of Malaysian Children (SEANUTS Malaysia) [9] that showed more than half the children and adolescents were classified as having low levels of PA and high levels of sedentary behaviour. These findings suggested the need for immediate and effective approaches to tackle this “inactivity epidemic.”

The strength of the present study is that the PA module was tailored to the needs of Malaysian adolescents who were O/O. The module was field-tested among potential participants and had its content validated by expert panel before being implemented in an intervention study. This enhanced the effectiveness of the module as content refinement was accomplished. One of the limitations in this study was the PA module was developed in only one language, i.e., Malay. Translating the PA module into English and other languages commonly used in Malaysia will minimize language barriers for adolescents attending various types of school with different mediums of instruction. Although the module was tailored for Malaysian adolescents, the principle can be applied or adopted for use by adolescents in other parts of the world. Another limitation was that validation of the PA module was done by only two education experts. However, once the content and face validity were done, we recruited secondary school teachers (*n* = 5), parents (*n* = 5) and health professionals (*n* = 5) to evaluate the acceptance of the module by using Tool to Evaluate Materials Used in Patient Education (TEMPtEd) questionnaire. The majority of the panel members (93%) agreed that the PA education module was generally suitable for use among adolescents. The results of the TEMPtEd evaluation had been reported by Wong et al. [56].

As students have hectic school lives, the CERGAS PA education programme was designed as a two-day intensive course conducted at a weekend residential camp to minimize the frequency of school visits by researchers and to ensure that the adolescents can focus on academic matters without too many disturbances during school hours. As an alternative to this intensive programme, we recommend future studies to test the effectiveness of employing the CERGAS PA module as part of the 12-week after-school sessions.

## 5. Conclusions

The PA module was developed specifically for use in the CERGAS, which is an obesity intervention programme targeting adolescents. The module was content validated and can be used to enhance PA knowledge among overweight and obese adolescents in Malaysia. The content of the CERGAS PA education module has the potential to be incorporated into school physical education textbooks to benefit more secondary school adolescents.

## Figures and Tables

**Figure 1 ijerph-16-01506-f001:**
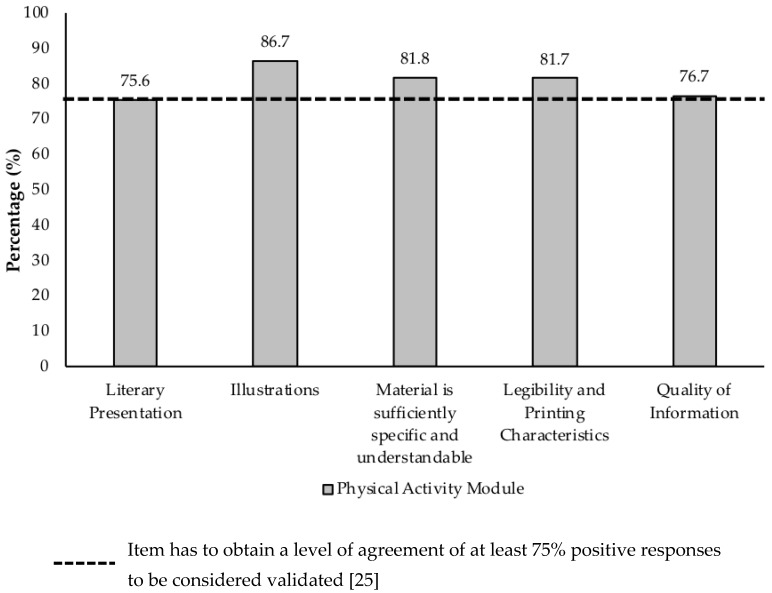
Level of agreement among experts for evaluative aspects of face validity.

**Figure 2 ijerph-16-01506-f002:**
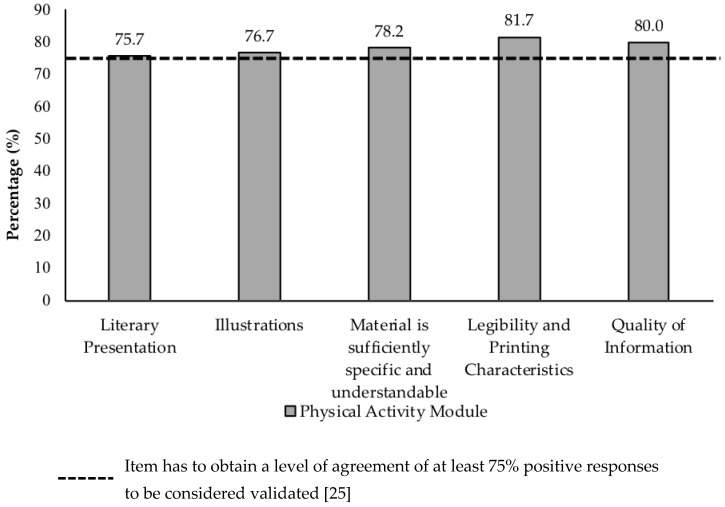
Level of agreement among adolescent target audience for evaluative aspects of face validity.

**Table 1 ijerph-16-01506-t001:** Characteristics of subjects *n* (%).

Demographic Characteristics	Boys (*n* = 25)	Girls (*n* = 19)	Overall (*n* = 44)
	*n* (%)	*n* (%)	*n* (%)
Age			
13 years old	8 (32.0)	7 (36.8)	15 (34.1)
14 years old	4 (16.0)	1 (5.3)	5 (11.4)
15 years old	13 (52.0)	11 (57.9)	24 (54.5)
Weight status			
Underweight	4 (16.0)	1 (5.3)	5 (11.4)
Normal weight	11 (44.0)	7 (36.8)	18 (40.9)
Overweight	5 (20.0)	3 (15.8)	8 (18.2)
Obese	5 (20.0)	8 (42.1)	13 (29.5)

**Table 2 ijerph-16-01506-t002:** Summary of main themes for qualitative findings.

Topics	Main Themes
PA/Exercise Topics of Interest	▪ Benefits of exercise ▪ Proper exercise techniques ▪ Ways to increase PA ▪ FITT (Frequency, Intensity, Time, Type) ▪ Ways to stay motivated to exercise regularly
Educational Materials Design	▪ Organization ▪ Layout and typography ▪ Content ▪ Illustration and cover ▪ Language

**Table 3 ijerph-16-01506-t003:** Topics, learning outcomes, content and activities of the Physical Activity (PA) module.

Unit	Topics	Learning Outcomes	Content	Activities	Learning Time
1	Let’s Be Active	Participants will learn to define PA. Participants will learn to explain the PA pyramid. Participants will explore benefits of PA. Participants will explore different levels (low, moderate and high) of PA intensity. Participants will learn to set PA goals.	Definition of PA The Importance of PA PA Pyramid PA Intensity Activity: PA Agreement Contract	- Participants complete “goal setting worksheet” with nutritionist/physical education expert/sport scientist. - Participants write down strategies to help to achieve goals. - Researcher/facilitators have face-to-face discussion with subjects to set realistic goals. Participants to discuss the barriers they are facing and create a coping strategy for each barrier.	60 min
2	Exercise and Fitness	Participants will learn to explain the principle of FITT (Frequency, Intensity, Time, Type). Participants will learn exercises that will develop strength and cardiovascular fitness. Participants will learn how to incorporate fitness activities into a healthy and active lifestyle. Participants will demonstrate proper form and techniques during exercise sessions. Participants will learn how to monitor their own heart rate.	Definition of Exercise Principle of FITT (Frequency, Intensity, Time, Type) Fitness Components Exercise to Improve Fitness Proper Exercise Techniques How to Monitor Heart Rate	- Hands-on activity: measure heart rate. - Certified exercise instructors demonstrate and guide participants to perform exercise correctly.	120 min
3	Staying Safe During Exercise and Physical Activity	Participants will learn skills to safely engage in physical activity. Participants will learn proper warm-up and cool-down exercise. Participants will learn safe use of equipment.	Proper Sports Attire Warm-Up and Cool-Down Exercises Water Intake Before, During and After Exercise Safety Issues During Exercise	- Certified exercise instructors demonstrate and guide participants to perform warm-up and cool-down exercise correctly.	90 min
4	Planning for Success	Participants will develop an exercise plan that will meet all fitness components. Participants will learn how to implement their individual exercise plan.	Introduction to Cross Training The Importance of Exercise Planning Tips to plan regular PA and exercise Let’s Get Started: Starting an Exercise Programme	- Participants need to plan their activity schedule.	60 min
5	How to Overcome Sedentariness and Get Moving	Participants will learn to define sedentary activity. Participants will explore different types of sedentary activities. Participants will develop a plan to reduce sedentary activity. Participants will learn personal skills and attitudes to increase PA level.	Definition of Sedentary Activity Health Implications of Sedentary Behaviour Ways to Reduce Activity Sedentary How to Increase PA Level	- Physical education expert and participants discuss together to find ways to reduce sedentary behaviour and overcome barriers. - Participants play games that requires them to move more. Tasks difficulty level is increased.	60 min

**Table 4 ijerph-16-01506-t004:** Content validity index for physical activity module by expert panel members (*n* = 5).

Criteria	Item Description	Relevant (Rating 3 or 4)	Not Relevant (Rating 1 or 2)	CVI ^a^	Interpretation ^b^
Scientific accuracy	Contents are in agreement with the current knowledge	4	1	0.80	Appropriate
	Recommendations are necessary and are correctly approached	4	1	0.80	Appropriate
Average CVI				0.80	
Content	Objectives are evident	5	0	1	Appropriate
	Recommendation about the desired behaviour is satisfactory	4	1	0.80	Appropriate
	There is no unnecessary information	5	0	1	Appropriate
	Important points are reviewed	4	1	0.80	Appropriate
Average CVI				0.90	
Overall CVI				0.85	Appropriate

^a^ CVI (Content Validity Index): the number of expert panel members who rated the item as relevant (rating 3 or 4) divided by the total number of expert panel members (*n* = 5); ^b^ CVI is higher than 0.79, the item is appropriate. If between 0.70 and 0.79, item needs revision. If less than 0.70, item is eliminated.

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
