# Peer review of "Development and Validation of a Physical Activity Educational Module for Overweight and Obese Adolescents: CERGAS Programme"

_ijerph, 2019, doi:10.3390/ijerph16091506_

Reviewer 1 Report

This is a well-written development and validation study, describing the development of an educational module to adolescents on physical activity and its importance on staying healthy and preventing disease. The need for the study and the material developed is well-defined in the introduction and the methodology used is sound. The development process is conducted and reported in three phases: needs assessment (I), development of the module (II) and content and face validation (III). All the phases have been performed rigorously by using justified methods, such as focus group discussions (Phase I) and content validity index (CVI). The authors have critically reflected all stages of the development process and reported the process precisely. The conclusion is justified. However, a few more specific definitions are needed to help the reader in putting the study in context.

Firstly, as the definition of adolescence in the literature varies, the definition or age range used in this study should be defined.

Secondly, it is not described in the manuscript (only in the conclusions) that where and how will the developed module be used or incorporated. This should be described in short.

Thirdly, the C.E.R.G.A.S Programme is mentioned already in the title, however the concept of the programme is not described in the manuscript.

A few minor comments:

Focus groups:

it is normal to conduct pilot discussions to test the pre-defined questions and their suitability. Was this done in the current study?

was there any other way provided for the adolescents to report their thoughts, f. ex. post-it notes? This is also quite common way to make sure that all the information is gathered.

Table 4:

in the columns reporting the n of the participants reporting the items as relevant or not relevant, perhaps the n should be included in the column heading to increase readability.

Author Response

Dear Editor,

We thank you and the reviewers for the constructive comments and thoughtful recommendations. Enclosed please find the revised version of our article Manuscript ID IJERPH-464835. Below are our point-by-point responses to the comments.

No.

Reviewer’s   Comments

Authors’   Response

1.

Firstly, as the definition of adolescence in the   literature varies, the definition or age range used in this study should be   defined.

Subjects were boys and girls aged 13 to 15   years old recruited from secondary schools.

We have now defined the age of the   adolescent subjects in Line 127.

2.

Secondly, it is not described in the manuscript (only in   the conclusions) that where and how will the developed module be used or   incorporated. This should be described in short.

C.E.R.G.A.S   participants will receive PA education in a two-day camp at a training   centre. This PA education module can serve as the main education material for   C.E.R.G.A.S’s participants and facilitators in the education camp.

We have added this   information at Lines 106-107.

3.

Thirdly, the C.E.R.G.A.S Programme is mentioned already   in the title, however the concept of the programme is not described in the   manuscript.

C.E.R.G.A.S is a 12-week programme, in   which participants receive additional nutrition and PA education delivered by   health professionals in a two-day camp, apart from the standard Physical and   Health Education modules offered in the school curriculum. Participants also took   part in exercise training sessions twice weekly for 12 weeks at school (90   minutes per session).

This information is included in Lines 62-68.

4.

It is normal to conduct pilot   discussions to test the pre-defined questions and their suitability. Was this   done in the current study?

The FGD interview protocol was reviewed by   expert panel comprising dietitian, nutritionist and qualitative research   expert. Suggestions from   expert panels were reviewed and amendments were done accordingly.

Lines 142-143.

Was there any other way provided for the   adolescents to report their thoughts, f. ex. post-it notes? This is also   quite common way to make sure that all the information is gathered.

Before   FGD sessions end, participants were given a chance to provide brief personal   commentary by writing on a piece of notes paper. Researcher   collected back the notes and incorporated these responses into module   preparation.

Lines 148-150.

5.

Table 4:

In the columns reporting the n of the   participants reporting the items as relevant or not relevant, perhaps the n   should be included in the column heading to increase readability.

The   total number of expert panel members is stated in the Title to Table 4.

Line   366

Reviewer 2 Report

Thank you for the opportunity to read and review your article. The topic of content, adolescent PA participation, is important. You have shown that you have undertaken a thorough process in the development of the educational tools. 

You present a sound summary of key literature that supports the need for inquiry into interventions that may increase adolescent participation in PA. However, the introduction section in the article lacks any discussion of the literature concerning the design and development of pedagogically sound intervention programmes in schools; the importance of integrating interventions into the core work of schools; and the need for partnership with education to develop and implement such interventions. It is important that you demonstrate in the introduction and follow through in the methods and discussion with more information about how and why partnership with the experts in the field (i.e. teachers) is established and how your programme designed supports the potential maintenance of this intervention as a sustainable programme for schools. 

You have presented key anthropometric evidence relevant to the study in the introduction but have omitted to describe relevant educational evidence. Please consider revising the introduction to include a section outlining the nature of the Health and Physical Education curriculum in Malaysian schools, available resourcing, learning objectives within the core curriculum with regard to PA/Exercise and national assessment evidence demonstrating understanding of the baseline you are working with (in the same way as you have provided this for the health aspects of your study (O/O rates etc). This will assist to enable readers to understand the context and implications of the evidence you present in the results section (e.g. section 3.2.1)

Finally, this is a paper about the design of an educational intervention, yet the introduction lacks any discussion of learning theory. Please outline the theoretical underpinnings of the pedagogical design that you have used and present an argument demonstrating why this design was considered appropriate to meet the stated aim of the intervention in the context of Malaysian schools.

In line 59 (and in the abstract) you refer to the C.E.,R.G.A.S programme. Could you please ensure that you define this acronym in line 50, and refrain from using the acronym (i.e. use the full name) in the abstract.

Your description of the methods that you have used is clear. 

Can you please clarify in the methods section who implemented the modules in this development period and who will be implementing the modules once they are validated. 

What period of class time is required for the intervention and how is this intended to be integrated into the core learning programmes of schools? 

Your intervention development committee is strongly weighted towards expertise in health. In section 2.3.1 we see that 2 of the 10 people are non-educators. In section 2.4.1 you state that your interdisciplinary panel consists of five people - one of whom was an educator. You then go on to describe a process of CVI which does not seem to account for the imbalance in the group. How did you address the fact that education expertise was under-represented in the scoring system that you used? This is a very important issue to address. If you did not address this, can you please include this in your limitations section and discuss the potential impact that this issue may have had on the evidence that you are presenting.  

Your reporting of the focus group evidence is sound. In your limitations section you may want to consider adding in a discussion regarding why you did not conduct FDGs with teachers in phase 1 and 2. 

Section 3.3 - Module content.

Could you please include in this section information about the expected learning time required to complete the units of work. 

In Table 3 you have defined the Learning Outcomes (LOs) for the module (i.e. all 5 units). It is most unusual that you have repeated these in each unit and have not identified how each unit contributes to the overall LOs with specific LOs. A more usual way to present a learning module overview would be to identify the specific LOs that are relevant to each unit. Learning involves the development of capabilities (knowledge, attitudes, skills and values) that are required to understand an issue or concept and potentially take action resulting from this understanding (i.e. developing capabilities associated with health literacy in the context of PA). Can you please expand table 3 to demonstrate more clearly the section you have labelled ‘content’ so as from an educational perspective readers are able to identify where attitudinal and value development is supported in these modules.  

Please add a limitations section following the discussion and address the potential implications of the limitations of your methods - in particular the lack of evidence from the education sector. 

Please conduct a final check of your article for minor grammatical errors. In particular you move between tenses in several places. 

Author Response

Dear Editor,

We thank you and the reviewers for the constructive comments and thoughtful recommendations. Enclosed please find the revised version of our article Manuscript ID IJERPH-464835. Below are our point-by-point responses to the comments.

1.

Lacks any discussion of   the literature concerning the design and development of pedagogically   sound intervention programmes in schools

Instead of school setting, the present   study employed residential education camp setting to improve adolescents’ PA   behaviors. The relevant discussion was added in introduction section.

Lines   62-82

2.

The importance of integrating   interventions into the core work of schools

The information is added in Lines 83-94.

3.

The need for partnership with education   to develop and implement such interventions.

The information is added in Lines 83-94.

4.

It is important that you demonstrate in   the introduction and follow through in the methods and discussion with more   information about how and why partnership with the experts in the field (i.e.   teachers) is established and how your programme designed supports   the potential maintenance of this intervention as a sustainable   programme for schools.

Instead of school setting, present study   employed residential education camp setting to improve adolescents’ PA   behaviors. The relevant discussion is now added in introduction section.

Lines   69-82

5.

Include a section outlining the   nature of the Health and Physical Education curriculum in Malaysian schools,   available resourcing, learning objectives within the core curriculum with   regard to PA/Exercise and national assessment evidence demonstrating   understanding of the baseline you are working with (in the same way as you   have provided this for the health aspects of your study (O/O rates etc).

The information is added in Lines 83-94.

6.

Lacks any discussion of learning   theory. Please outline the theoretical underpinnings of the   pedagogical design that you have used and present an   argument demonstrating why this design   was considered appropriate to meet the stated aim of the   intervention in the context of Malaysian schools.

Social Cognitive Theory was used.

The information is added in Lines   191-198.

7.

In line 59 (and in the abstract) you refer to the   C.E,R.G.A.S programme. Could you please ensure that you define   this acronym in line 50, and refrain from using the acronym (i.e. use   the full name) in the abstract.

C.E.R.G.A.S is the Malay acronym of CEria (cheerful), Respek   (respect), Gigih (persistent), Aktif (active) and Sihat (healthy).

Lines 25-26

Lines 60-61

8.

Can you please clarify in the methods section   who implemented the modules in this development period and who will   be implementing the modules once they are validated. 

The information is added in Line 116-121.

9.

What period of class time is required for the intervention   and how is this intended to be integrated into the core learning programmes   of schools? 

Participants received additional PA education in a two-day   residential camp. This was an intensive education camp and conducted in   weekend (Saturday and Sunday) by considering budgetary, time and resources   constraints.

In view of the hectic school life, PA education camp which   was designed for a two days course covering at least 15 hours of training was   considered adequate (Klohe-Lehman et al. 2006; Kabahenda et al. 2011; Inayati   et al. 2012). Moreover, researchers can minimize the visit frequency to   school to ensure that the school adolescents focusing on academic matters   without too many disturbances. Furthermore, intensive education programme   might have positive implication as the participants could fully concentrate   on learning process without distractions. In addition, the participants can   apply the knowledge gained into their daily life immediately. Hence, our   C.E.R.G.A.S modules were indicated for two days training.

Lines 72-82

Once the effectiveness is proven, the contents of   C.E.R.G.A.S PA education module can be inserted in school physical education   (PE) textbook to improve students’ PA knowledge, attitude and practice.

Lines 116-121

10.

Your   intervention development committee is strongly weighted towards expertise in   health. In section 2.3.1 we see that 2 of the 10 people are non-educators. In   section 2.4.1 you state that your interdisciplinary panel consists of   five people - one of whom was an educator. You then go on to describe a   process of CVI which does not seem to account for the imbalance in the group.   How did you address the fact that education expertise was under-represented   in the scoring system that you used? This is a very important issue   to address. If you did not address this, can you please include this in   your limitations section and discuss the potential impact that this issue may   have had on the evidence that you are presenting.  

We acknowledge that the PA module was reviewed by only two   education experts, and have noted this in our Limitation

Lines 482-488.

11.

Your   reporting of the focus group evidence is sound. In your limitations section   you may want to consider adding in a discussion regarding why you   did not conduct FDGs with teachers in phase 1 and 2. 

Instead   of putting this as limitation, we explained in the method section why we   conduct FGDs with adolescents only.

C.E.R.G.A.S   programme is using “adolescent-centred” approach, thus, FGD was employed to   explore adolescents’ opinions, ideas, perceptions and concerns in regard to   education contents, layout and design.

Line 123

12.

Section 3.3

Could you please include in this section information   about the expected learning time required to complete the units of   work. 

We   have now added learning time in column 6 of Table 3.

Line 347

13.

Section 3.3

In Table 3 you have defined the Learning Outcomes (LOs)   for the module (i.e. all 5 units). It is most unusual that you have repeated   these in each unit and have not identified how each   unit contributes to the overall LOs with specific LOs. A more usual way   to present a learning module overview would be to identify the specific LOs   that are relevant to each unit. Learning involves the development of   capabilities (knowledge, attitudes, skills and values) that are required   to understand an issue or concept and potentially take action resulting from   this understanding (i.e. developing capabilities associated with health   literacy in the context of PA). Can you please expand table 3   to demonstrate more clearly the section you have labelled ‘content’   so as from an educational perspective readers are able to identify where   attitudinal and value development is supported in these modules.  

We   have now added learning outcomes for each topic in Table 3.

Line 347

Round  2

Reviewer 2 Report

Thank you for the consideration you have given to the review comments. 

The addition of the explanation of CERGAS helps enormously to place the project into context. 

In the final sentence of the introduction you state that the PA education module will serve as the main education material in the two-day camp. Can you please clarify as to whether or not you will also employ the 12-week after-school sessions in the future. These after-school sessions would theoretically be very important (and maybe you should test this out) as it is well known that a burst of training or education (such as a two-day camp) without suitable ongoing support is unlikely to produce long-term change in behaviours. 

Can you please check your grammar in lines 115 - 120. Did you intend to state they the validates PA module would serve as reference material in the future? In the last sentence you imply that before you undertook this trial you validated the module? Was that not what you were doing in this trial. I suspect this is simply a case of the sense of the sentence being incorrect? This needs attention as it is very confusing. 

Please start line 122 with The......

Thank you for outlining your use of Social Cognitive Theory. As an educator I would be interested in knowing more about your pedagogical approach- but I appreciate that you have made an effort to explain something of the theory that should be underpinning your work. 

Thank you for adding in the detail of the Learning Outcomes. It is important that you are realistic in your statements. You cannot actually be sure that at the end of the learning module all students WILL be able to do everything you have listed. In educational literature it is more common to use language that describe what students will do - as until you assess the learning you cannot know that you will get 100% achievement of the expected outcome. For example, you might re-phrase your LOs as:

Participants will

examine different types of physical activity and develop understanding of factors  considered essential in a definition of PA

explore the components of physical activity pyramid 

explore differences between the types of activities that are categorised as low, medium or high PA

learn about goal setting in relation to PA and experience setting their own PA goals

These represent learning objectives - which you should be measuring to determine the learning outcomes for each student. If you do not change this, could you at least phrase it as expected learning outcomes so that it is clear that you understand that you cannot confirm outcomes until you measure these, and you will see heterogeneity in the outcomes across any class or cohort. 

Thank you for addressing the limitation relating to the lack of educational input into the design of the study, 

Could you please review the manuscript for grammatical errors. These are minor, and a reader can work with those and understand the work. However, they do detract from the content of your manuscript and make it less accessible. 

I look forward to seeing how this work develops in the future and hope you are able to continue with the work and publish the outcomes of the wider implementation phase. 

Author Response

Dear reviewer, 

We thank you for the constructive comments and thoughtful recommendations. Enclosed please find the revised version (Round 2) of our article Manuscript ID IJERPH-464835. Below are our point-by-point responses to the comments (please see attachment).

No.

Reviewer’s   Comments

Authors’   Response

1

In   the final sentence of the introduction you state that the PA education module   will serve as the main education material in the two-day camp. Can you please   clarify as to whether or not you will also employ the 12-week after-school   sessions in the future. These after-school sessions would theoretically be   very important (and maybe you should test this out) as it is well known that   a burst of training or education (such as a two-day camp) without suitable   ongoing support is unlikely to produce long-term change in behaviours. 

We   have added this as recommendation in Lines 483-488.

As   an alternative to this intensive programme, we recommend future studies to   test the effectiveness of the 12-week after-school sessions employing   C.E.R.G.A.S PA module.

2

Please start line 122 with   The......

We have amended accordingly.

Line 120

3

Thank you for outlining your   use of Social Cognitive Theory. As an educator I would be interested in   knowing more about your pedagogical approach- but I appreciate that you have   made an effort to explain something of the theory that should be underpinning   your work. 

We   have added the information in Lines 198-202.

The pedagogical approach we used in   present study was constructivism teaching. This method of teaching helps   participants to better relate the information learned in the education camp   to their lives. In a constructivism classroom, participants work in groups.   This helps participants learn social skills, support each other’s learning   process and value each other opinion and input.

4

Thank you for adding in the   detail of the Learning Outcomes. It is important that you are realistic in   your statements. You cannot actually be sure that at the end of the learning   module all students WILL be able to do everything you have listed. In educational   literature it is more common to use language that describe what students will   do - as until you assess the learning you cannot know that you will get 100%   achievement of the expected outcome. For example, you might re-phrase your   LOs as:

Participants will

examine different types of   physical activity and develop understanding of factors  considered   essential in a definition of PA

explore the components of   physical activity pyramid 

explore differences between   the types of activities that are categorised as low, medium or high PA

learn about goal setting in   relation to PA and experience setting their own PA goals

These represent learning objectives - which   you should be measuring to determine the learning outcomes for each student.   If you do not change this, could you at least phrase it as expected learning   outcomes so that it is clear that you understand that you cannot confirm   outcomes until you measure these, and you will see heterogeneity in the   outcomes across any class or cohort. 

We have now revised   each Learning Outcome to reflect more what the student will learn.

Line 343

5

Could you please review the manuscript for   grammatical errors. These are minor, and a reader can work with those and   understand the work. However, they do detract from the content of your   manuscript and make it less accessible. 

We have reviewed   for English grammar accordingly.
